# The Role of Cutinsomes in Plant Cuticle Formation

**DOI:** 10.3390/cells9081778

**Published:** 2020-07-25

**Authors:** Dariusz Stępiński, Maria Kwiatkowska, Agnieszka Wojtczak, Justyna Teresa Polit, Eva Domínguez, Antonio Heredia, Katarzyna Popłońska

**Affiliations:** 1Faculty of Biology and Environmental Protection, Institute of Experimental Biology, Department of Cytophysiology, University of Lodz, Pomorska 141/143, 90-236 Lodz, Poland; dariusz.stepinski@biol.uni.lodz.pl (D.S.); maria.kwiatkowska@biol.uni.lodz.pl (M.K.); agnieszka.wojtczak@biol.uni.lodz.pl (A.W.); justyna.polit@biol.uni.lodz.pl (J.T.P.); 2Instituto de Hortofruticultura Subtropical y Mediterránea “La Mayora” UMA-CSIC, Universidad de Málaga, Campus de Teatinos, 29071 Málaga, Spain; edominguez@eelm.csic.es (E.D.); heredia@uma.es (A.H.)

**Keywords:** cuticle-synthesizing enzyme, cutin, cutinsome, electron microscopy, plant cuticle

## Abstract

The cuticle commonly appears as a continuous lipophilic layer located at the outer epidermal cell walls of land plants. Cutin and waxes are its main components. Two methods for cutin synthesis are considered in plants. One that is based on enzymatic biosynthesis, in which cutin synthase (CUS) is involved, is well-known and commonly accepted. The other assumes the participation of specific nanostructures, cutinsomes, which are formed in physicochemical self-assembly processes from cutin precursors without enzyme involvement. Cutinsomes are formed in ground cytoplasm or, in some species, in specific cytoplasmic domains, lipotubuloid metabolons (LMs), and are most probably translocated via microtubules toward the cuticle-covered cell wall. Cutinsomes may additionally serve as platforms transporting cuticular enzymes. Presumably, cutinsomes enrich the cuticle in branched and cross-linked esterified polyhydroxy fatty acid oligomers, while CUS1 can provide both linear chains and branching cutin oligomers. These two systems of cuticle formation seem to co-operate on the surface of aboveground organs, as well as in the embryo and seed coat epidermis. This review focuses on the role that cutinsomes play in cuticle biosynthesis in *S. lycopersicum, O. umbellatum* and *A. thaliana*, which have been studied so far; however, these nanoparticles may be commonly involved in this process in different plants.

## 1. Introduction

The cuticle, which mainly covers the epidermis of the aerial parts of terrestrial plants, forms a physical barrier between a plant and its environment, and thus it constitutes a very important biological element. By protecting plants against excessive evaporation, it has allowed them to leave the water habitat and colonize land. In some species, the cuticle forms highly repellent leaf surfaces that have the ability to self-clean [1,2], which provides a maximum capacity for photosynthesis [3,4]. The mechanical features of the cuticle, and thus its protective properties, depend at least partly on its thickness on fruit and leaves of different species and on the amounts of its particular components [5,6,7,8]. Moreover, in some consumable species, the cuticle thickness was reported to influence fruit cracking due to changes in the mechanical resistance to deformation [9,10,11,12]. Although the cuticle seems to be quite a resistant layer, pathogenic injuries, especially fungal infections, may cause defects in its structure [13,14,15]. Notwithstanding a great number of papers stimulated by the abundance of cuticles in nature and their great economic importance, there are still many important issues regarding cuticles that need elucidation, especially since it significantly contributes to the quality and shelf-life of plant food products. One of the interesting aspects still requiring deep analysis is the polymerization of cutin from its chemically modified monomers synthesized in epidermal cells [7,16,17,18,19,20].

The cuticle consists of an amorphous, viscoelastic cutin, which is insoluble in organic solvents, and of intracuticular and epicuticular waxes soluble in these solvents. Cutin is ubiquitous in terrestrial plants and is characterized by specific physicochemical features [21]. It is composed of esterified bi- and trifunctional fatty acids (C16 and C18). The other components, waxes, contain flavonoids, sterols and triterpenoids, as well as a mixture of very long chain fatty acids (VLCFAs, C20 to C34) [22,23]. Biosynthesis pathways of cutin and wax use common precursors and mechanisms of translocation, and are regulated through similar networks of transcription factors [8,24,25,26]. Some of them belonging to the homeodomain leucine zipper IV (HD-ZIP IV) family, which are expressed in the epidermis, were shown to modify cuticle deposition as well as epidermal properties [27,28]. These fatty acid precursors are transformed into cutin and wax monomers. Members of the ABCG (G subfamily of Adenosine Triphosphate (ATP)-binding cassette) transporters were reported to be involved in their transport to the outer surface [3,29,30,31]. Lipid transfer proteins (LTPG) were also reported to be involved in the export of cuticular components [32,33,34], but the mechanism underlying their roles in this process remains to be determined.

Recently, many reviews have appeared, which have consolidated knowledge on the enzymatic pathway of cuticle lipid component biosynthesis, cutin and waxes [3,7,17,18,19,35,36,37,38]. These works presented cuticle biosynthesis from different angles and were based on various experimental models, including mutant analysis and in vivo labeling. The current review is devoted to the non-enzymatic pathway of cutin biosynthesis, which has recently been described in the literature, and only necessary information on enzymatic cutin biosynthesis is included. Fatty acids synthesized de novo in plastids are cutin precursors which are transported to the endoplasmic reticulum (ER), where they are transformed into monoacyl- and diacylglycerols (MAG, DAG) by glycerol-3-phosphatase-CoA acyltransferase (GPAT6) and diacylglycerol acyltransferases (DGAT2), respectively [39,40]. Some of these lipids form lipid bodies between two leaflets of ER membrane [41,42]. These lipid bodies then separate from ER, grow as a result of lipid synthesis with the involvement of GPAT6 and DGAT2, and maturate in the cytoplasm [30,42,43,44,45]. Moreover, the lipids are hydrolyzed by phospholipase D and lipase located at the surface of these structures [42,45]. The latter enzyme could also polymerize lipids according to the idea proffered by Gómez-Patiño et al. [46]. The products of these reactions are transported from lipid bodies to the cell wall via microtubules. Lipid monomers from ER and from lipid bodies separate into two independent pathways through which they are transported to the cell wall to form the cutin matrix of the cuticle [42,43,45,47] (Figure 1).

To date, a large number of enzymes involved in the biosynthesis of cutin have been identified [8,17,19,38,48]. Some of them, mainly the abovementioned glycerol acyl transferases, are involved in the biosynthesis, activation and modification of fatty acids [38], whereas others seem to participate in cutin monomer polymerization. Among the latter enzymes, there are DEFECTIVE IN CUTICULAR RIDGES (DCR) cytoplasmic acyltransferase, which belongs to the BAHD family [49,50], as well as BODYGUARD (BDG) [51,52] and CUTIN SYNTHASE 1 (CUS1) [53,54], both extracellular enzymes belonging to the α/β superfamily of hydrolases. The involvement of CUS1 in tomato fruit cutin polymerization was extensively studied [48,53,54,55,56]. The enzyme was localized in the fruit cuticular layer by means of antibodies, and its role in cuticle synthesis was revealed after the analysis of tomato mutants and the activity of CUS1 recombinant enzyme [53,54,57]. Thus, cutin polymerization takes place at the site of its deposition, i.e., in the growing cuticle. Moreover, a homologous enzyme, CUS2, was shown to be involved in the maintenance of *Arabidopsis thaliana* sepal cuticular ridges [58].

The other mechanism of cutin formation is based on the involvement of specific structures, called cutinsomes, in this process. In many species, electron-dense globular nanostructures were reported to be in close contact with the inner side of the growing cuticle at early stages of development, and they were suggested to be cutinsome-like particles [35,59,60,61]. An in vitro study revealed that dihydroxy fatty acids, one of the main cutin monomers, were able to form spherical or granular nanostructures [46,62,63,64,65]. These structures were formed via physicochemical processes, based on the ability of hydroxy fatty acids to self-assemble and self-esterify spontaneously into nanoparticles of 40–200 nm in diameter, named cutinsomes [62,63,64,65]. They are composed of two structurally and chemically different parts. Their cores are filled with branched and cross-linked hydroxy acids, while the outer parts are hydrophilic carboxylate/carboxylic (-COO/-COOH) shells separating the lipid, liquid-like content from the aqueous environment [5,66,67]. Moreover, the shell also contains pectins [68]. Cutinsomes are capable of spontaneous in vitro polymerization forming an amorphous cutin-like film. Since in vitro cutinsomes were composed of naturally occurring cutin monomers, it was postulated that they could be present *in planta* and participate in cutin synthesis [63,64]; hence, antibodies recognizing these structures were generated [66]. Cutinsomes were immunolocalized in the ground cytoplasm or specific cytoplasm domain (lipotubuloid metabolon—LM, described in Section 2.2), as well as in close proximity to the growing cuticle of several species [30,31,66,69]. The possibility of a non-enzymatic pathway of cutin biosynthesis was considered by other researchers [38,48].

This review focuses on the available information on cutinsome participation in cuticle biosynthesis in three angiosperm species, *Solanum lycopersicum* [66], *Ornithogalum umbellatum* [69] and *Arabidopsis thaliana* [31], in which these structures were identified with the use of the TEM immunogold method and anti-cutinsome antibodies, which only recognize cutinsomes while other lipid components and cell wall elements are not labeled [30,32,66,69]. The involvement of cutinsomes in the cuticle formation of other plant species is also considered.

## 2. Cutinsomes of Different Plant Species

### 2.1. Solanum Lycopersicum Cutinsomes

*In planta* ultrastructural studies of *S. lycopersicum* fruit epidermis (Figure 2A–D) showed cutinsomes and particles resembling cutinsomes merging into a procuticule at the early stage of cuticle formation (five to ten days post anthesis) [7,66,70]. However, ten days post anthesis, CUS1—which was not present at earlier developmental stages—was observed [54]. This coincided with a dynamic thickening of the cuticle [70]. These results suggest that the way in which cutin forms may depend i.a., on the stage of organ development [18,71]. Indeed, it has recently been shown that CUS1 is not involved in the synthesis of the procuticle but that there is a coordinated and temporal sequence that involves cutinsomes contributing to the synthesis of the procuticle and CUS1 participating later during the development in the further incorporation of cutin material to the procuticle template [56]. Thus, these two theories are not mutually exclusive but rather complementary. This is probable, since oligomers or oligoesters, as well as 2-MAG, could be substrates for CUS1 [37], which is capable of forming both branched [19] and linear polymers in vivo, or small oligomers in an aqueous environment in vitro [54,55], while branched and cross-linked oligomers are also synthesized and delivered by cutinsomes [18,71]. Highly branched fatty acids can be expected in tomato and pumpkin, as dihydroxy fatty acids account for more than 80% of the cutin fatty acids ((9/10)-16 hexadecanoic fatty acid) [72]. Thus, the hypothesis that cutinsomes are the source of these polymers seems highly likely, since these nanostructures contain them [5,66,67].

Ultrastructural observations of a tomato mature embryo (Figure 3A) revealed cutinsome-like structures in the cotyledon epidermis (Figure 3B) and the first cell layer of the radicle cap (Figure 3C). The external cell walls of these organs were covered with a prominent cuticle (Figure 3B,D). Interestingly, a root cap cuticle was observed for the first time in tomato lateral roots and in *Arabidopsis* primary roots [73]. Thus, it cannot be ruled out that these cuticles were formed at least partially due to cutinsome involvement.

A model of cuticle formation in different tomato organs, involving cutinsome and CUS1 and based on TEM immunogold studies, is presented in Figure 4 [17,30,54,55,66].

### 2.2. Ornithogalum Umbellatum Cutinsomes

*O. umbellatum* ovary epidermis (Figure 5A,B) is characterized by the presence of LM (Figure 5B), a specific cytoplasm domain, mainly consisting of lipid bodies entwined with a system of microtubules running in different directions, which are bound with actin filaments by myosin and kinesin [44]; hence, LMs are capable of autonomic motility, including rotation and linear motion [74]. As these structures are part of the cytoplasm, they are also rich in free ribosomes, polysomes and ER (Figure 6). Individual mitochondria, Golgi structures, microbodies, as well as autolytic vacuoles at later developmental stages of the ovary epidermis, can also be observed in them. LMs appear in young organs, and, when the cells reach their final size (approximately 30 times bigger than the initial stage), they break into single lipid bodies. LMs are involved in the synthesis of lipids and cuticle lipid components [43]. Their role in the synthesis of the latter was proved by autoradiography [69].

In dynamically growing epidermal cells of the ovary, cutinsomes are present in different areas of LMs, mainly near lipid bodies, ER and microtubules, i.e., at the sites where cutin precursors can be found (Figure 6) [44]. In TEM, the greatest numbers of cutinsomes are visible near microtubules, both in the sections without and with the cutinsome antibodies (Figure 5D,E) [30,69]. Such a localization of cutinsomes allows one to assume that microtubules transport them to the outer cell wall. The analysis of numerous TEM images suggests that cutinsomes leave LMs, the site of their formation, cross the plasmalemma and the polysaccharide wall (Figure 5F,G), and then reach the cuticle (Figure 5C and Figure 6) [69], analogously to the translocation of radioactive palmitic acid in autoradiograms labeled with silver particles [43]. It seems possible that the cutinsomes are able to go through hydrophilic cytoplasm due to the hydrophilic shell covering their hydrophobic core [65].

The latest research suggests that cutinsomes are also platforms enabling the transport of GPAT6 and WS/DGAT2 from the cytoplasm to cuticle (Figure 6). The enzymes GPAT6 and WS/DGAT2, together with LTPG, are involved in the synthesis of cutin and waxes, respectively [30,32,75].

### 2.3. Arabidopsis Thaliana Cutinsomes

The cuticles of certain organs of *Arabidopsis* differs from those found in most plant species. Moreover, these various organs are covered with different cuticles. For example, the cuticle of leaves and stems [76] has ultrastructural features that are different from those in petals [77]. The features of the former organs probably result from an atypical leaf cuticle composition, which is characterized by the prevalence of unsaturated and dicarboxylic fatty acids with reduced amounts of *ω*-hydroxy fatty acids [78], whereas other organs, such as petals for example, have a regular cuticle composition [49]. Similarly, an uncommon cuticle was found in soybean seed coat cuticle [79].

*A. thaliana* seeds have multiple tissues covered with epidermis that has the capacity to deposit cutin with some modifications, which presumably significantly influence the development of the embryo and seed [80]. The cutin matrix partially merges with the cell wall components, mainly polysaccharides and pectins, forming an amorphous structure during the early stages of embryo development [36]. In the case of the cuticle of the *A. thaliana* embryo, it is present from the globular stage up to the mature embryo stage (Figure 7A), as was observed after auramine O staining [81]. Even though De Giorgi et al. [82] only noticed a slightly visible embryo cuticle via TEM, Yang et al. [83] and Stępiński et al. [31] showed the presence of a distinct embryo cotyledon cuticle (Figure 7B,D). In addition, the endosperm cuticle was observed at the mature seed stage [82].

In TEM, cutinsomes participating in cuticle synthesis were first observed in *A. thaliana* in the cotyledon epidermis of a mature embryo, and they were visible as dark, spherical, clearly delimited structures, about 40 nm in diameter (Figure 7C,D). In addition, the cutinsomes were also present as less visible structures, due to a different fixation mode, but they were labeled with antibodies (Figure 7D,E). They were visible in cytoplasm both in its central part (Figure 7C), as well as near the external cell wall (Figure 7B,D,E), in this cell wall and near cuticle, where cutinsomes underwent fusion (Figure 7E). In addition, Loubéry et al. [87] also suggest that cutinsomes could be involved in the formation of the endosperm cuticle of *A. thaliana* seeds, since electron-dense particles resembling cutinsomes were observed in TEM images of the cell walls with inner cuticle of wild-type seeds at the preglobular, heart and walking stick stages.

It cannot be excluded that cuticle with both a typical and atypical composition is formed with the participation of cutinsomes, which may provide basic cuticle components, whereas the differentiation of cuticle composition in particular organs is regulated during plant development.

## 3. Conclusions and Perspectives

The cutinsome mechanism is a relatively novel idea in plant cuticle creation, which most likely cooperates with the well-grounded enzymatic pathway. Studies on *cus1* mutants [48] showed the deposition of a normal procuticle, as well as the presence of cutinsomes at early stages of development [56]. Changes in cutin deposition were only observed in the *cus1* mutant after the cell division period; however, the lack of correlation between cutin deposition and *CUS1* expression during the cell expansion period suggests the participation of other enzymes [19,38,48,57].

To date, cutinsomes have been identified with anti-cutinsome antibodies *in planta* in three species presented in this review, although they can also be observed with the use of conventional TEM without antibodies. Even though researchers exploring cuticles noted electron-dark or osmiophilic granules near or/and in external cell walls, they did not recognize them as cutinsomes. Interestingly, the dark globular structures agglomerating at the internal side of the forming cuticle membrane were already observed in mosses [61]. In parasitic *Cuscuta gronovii*, as well as in *Utricularia sandersonii,* an insectivorous plant, procuticle transforming into cuticle proper was visible on the earliest protodermal areas. Globular electron-osmiophilic structures were revealed in the matrix at the transitional stage between the procuticle and cuticle proper [60]. It is tempting to say that these structures are cutinsomes. Particles similar to those in *U. sandersonii* were also present in the electron-dense procuticle-like layer of *Sphagnum fimbriatum* early protonema [61]. The studies on the adaxial side of the *Clivia miniata* (Amaryllidaceae, monocotyledonous) leaf with its cuticular membrane, where electron-dense globules merged below the cuticle proper, seem to provide further proofs of the presence of supposed cutinsomes [59]. Globular cutin-rich cystoliths coalescing to form a boundary between the cuticular layer and the cell wall were also observed in *Clivia* leaf [35,88]. Similarly, numerous spherical dark structures appear in the aerial cell wall of poplar leaf epidermis [85]. These nanoparticles seem to fuse to form the cuticle layer, as in tomato fruit epidermis (compare Figure 2D, Figure 4 and the scheme in Domίnguez et al. [66]). It is also worth noting that recent TEM studies of nectaries in *Epidendrum* [89], *Geranium* [90] and *Prunus laurocerasus* [91] flowers may point to the presence of cutinsomes, judging from the dark spherical structures located in the epidermal cell wall near the cuticle. Moreover, cutinsome-like structures were observed in the cell walls of the outermost endosperm layer of *Olea europea* 22 and 26 weeks after flowering [92].

It has also been recently discovered that in another plant coat, suberin from cork, there are particles named suberinsomes containing suberin monomers, which are assembled into crosslinked and branched components. These spherical bodies of 200–400 nm in diameter, which were compared to cutinsomes, are responsible for building large suberin aggregates and creating a suberin layer [93]. The process of these suberinsomes’ in vitro formation resembles that of cutinsome creation. It is also worth noting that self-assembly processes were also implicated in exine development in more than 30 plant species from a wide variety of taxa [94,95,96,97,98,99,100,101,102,103,104,105]. Moreover, *GELP77*, a GDSL-type esterase/lipase gene that is mainly expressed in *Arabidopsis* microspores and tapetal cells [106] at early stages of flower development [107], has recently been shown to participate in pollen wall development and has been suggested to play a similar function in pollen to that of CUS1 in *S. lycopersicum* fruit cuticle [106]. Hence, the coexistence of self-assembly and enzymatic processes was also suggested for pollen wall development. Taking the above into consideration, it seems that a non-enzymatic mechanism that participates in the protection of the outer surface of epidermal cell walls with fatty acid-derived polymers is a widespread phenomenon in the plant world.

Cutinsomes can be found in ovary, fruit, seed and mature embryo cotyledon epidermis, i.e., always when aerial, endosperm or embryonic cuticle is formed, and thus it is highly probable that they are the source of lipid cutin components, namely branched and cross-linked hydroxy acids, during cuticle deposition [5,66,67]. In the epidermis of species with LMs (e.g., *O. umbellatum*), cutinsomes are generated in them, whereas in that of species without LMs (e.g., *A. thaliana)*, they are only generated in ground cytoplasm. Preliminary studies of tomato young fruit epidermis allow one to suppose that it might contain structures resembling lipotubuloids; however, ultrastructural studies are needed to confirm this.

Although great progress has recently been made in cuticle formation studies, there are many issues that need to be addressed. For example, it is still not clear how the cutin matrix is linked with the polysaccharide cell wall components. Specific proteins were proposed to connect these two components [108]. It is not known how cutin elements present in cutinsomes are delivered to the cuticle. As the transport of cutinsomes through cytoplasm most probably proceeds via microtubules, it remains to be uncovered how cutinsomes are translocated through the cell wall. Another issue worth investigating is the transport of GPAT6 along with cutinsomes from the cytoplasm to the cuticular area [30], ensuring the synthesis of MAG, which is the substrate for CUS, in the region of cuticle formation [54]. GPAT6 and MAG are located near each other, as in a metabolon, suggesting their improved cooperation. As CUS-like enzymes are most probably present in dicotyledons, monocotyledons, gymnosperms, lycopodine and bryophyta [55], it would also be worth investigating if cutinsomes participate in cuticle biogenesis in different plant groups, including charophycean algae [109,110,111,112,113]. Interestingly, *Klebsormidium flaccidum*, a charophycean algae, is covered with a hydrophobic cuticle-like layer containing lipids and glycoproteins [111,112]. Although this layer is different from the cuticle present in land plants, homologs to ABCG transporters responsible for wax transport in *A. thaliana* were found in the *K. flaccidum* genome. Moreover, homologs of CUS and BDG were found in algae such as *Klebsormidium nitens*, *Penium margaritaceum* and *Mesotaenium endlicherianum* [114]. Although there is no evidence of true cutin or suberin in charophytes, wax-like lipid deposits were observed in the cell walls of *K. nitens* [111].

Cuticle genesis with the participation of cutinsomes cannot only be considered and comprehensively understood on the basis of the small number of research subjects described above. However, a highly probable hypothesis can be put forward, according to which cutinsomes are also present and also take part in cuticle formation in other angiosperm plants. Due to the fact that knowledge on the variety of plant organ coats is still scarce, research towards a better understanding of cutin, suberin and the possible coexistence of different mechanisms for their assembly is of particular importance.

## Figures and Tables

**Figure 1 cells-09-01778-f001:**
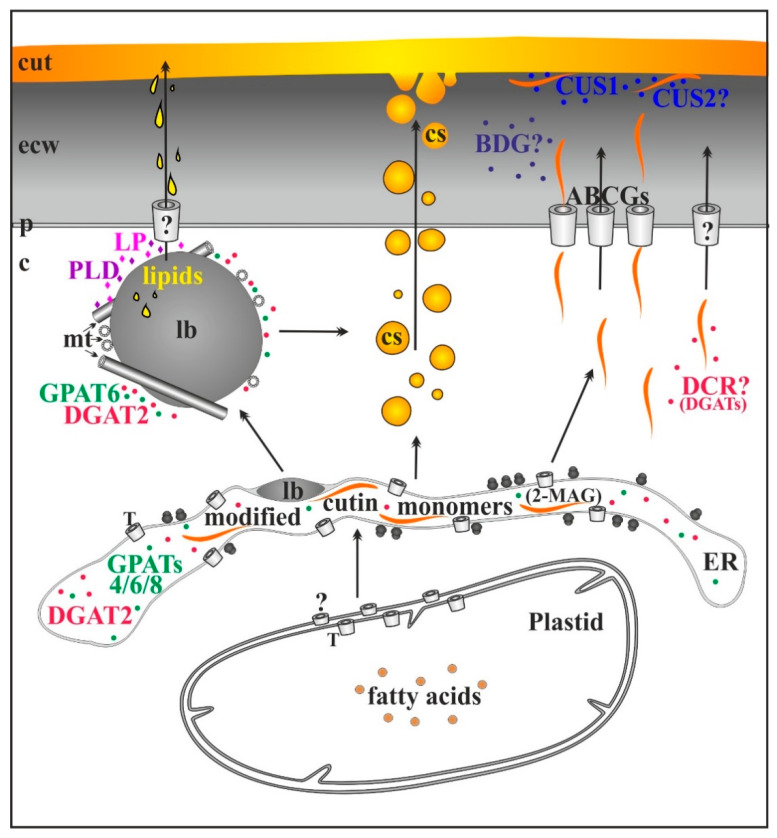
Scheme of complementary mechanisms of cutin biosynthesis by enzymes and cutinsomes. ATP-binding cassette transporters of the G subfamily (ABCGs); c, cytoplasm; cut, cuticle; cs, cutinsome bordered with a shell (black ring) and without it; enzymes participating in cutin biosynthesis: BODYGUARD (BDG), CUTIN SYNTHASEs (CUSs), DEFECTIVE IN CUTICULAR RIDGES (DCR), diacylglycerol acyltransferases (DGAT2) and glycerol-3-phosphatase-CoA acyltransferases (GPATs); ecw, external cell wall; ER, endoplasmic reticulum; lb, lipid body; LP, lipase; 2-monoacyl glycerol (2-MAG); mt, microtubule; p, plasmalemma; PLD, phospholipase D; T, transporters; question marks, possible involvement.

**Figure 2 cells-09-01778-f002:**
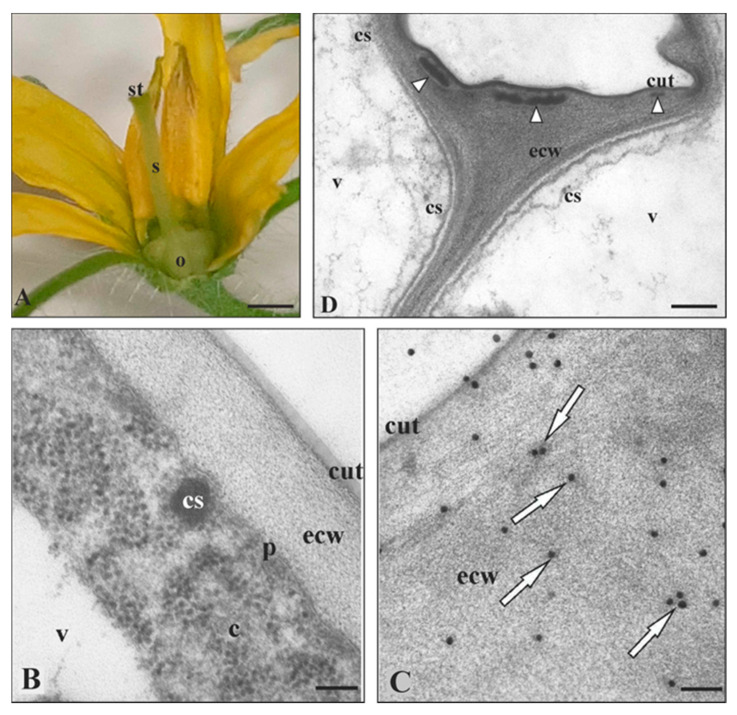
Cutinsome localization and identification in ovary of *S. lycopersicum*. (**A**) Mature flower with clearly visible ovary five days after anthesis. (**B**–**D**, TEM images) *In planta* cutinsomes in cytoplasm and external cell wall of fruit epidermis, more or less visible depending on the fixation method. Preparations fixed in glutaraldehyde with (**B**,**D**) OsO_4_ and (**C**) without OsO_4_. (**B**) Cutinsome with electron-dense contents near plasmalemma heading for external cell wall with clearly visible cuticle and (**D**) coalesced cutinsomes (arrow heads) forming cuticle layer. (**C**, arrows) Cutinsomes after immunogold reaction with anti-cutinsome antibodies labeled with one to a few 20 nm gold grains. Single grains localize cutinsomes, although these structures are not clearly visible on the external cell wall; c, cytoplasm; cs, cutinsome; cut, cuticle; ecw, external cell wall; o, ovary; p, plasmalemma; s, style; st, stigma; v, vacuole; Bars, (**A**) 700 µm, (**B**,**C**) 100 nm, (**D**) 200 nm.

**Figure 3 cells-09-01778-f003:**
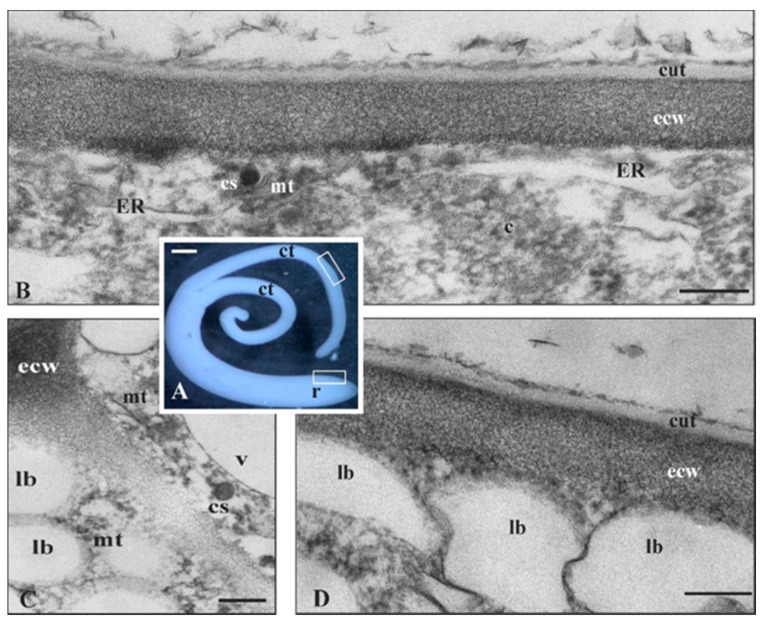
Cutinsomes in cuticle formation of *S. lycopersicum* embryo. (**A**) Mature embryo (frames show sites of studied material containing clear cuticle). TEM images of (**B**) cotyledon and (**C**,**D**) the radicle cup. Cutinsomes in (**B**,**C**) cytoplasm, near (**B**,**C**) plasmalemma travelling toward (**B**,**D**) the external cell wall in order to build a distinct layer of cuticle. (**B**–**D**) Material fixed in glutaraldehyde without OsO_4_; c, cytoplasm; ct, cotyledons; cs, cutinsome; cut, cuticle; ecw, external cell wall; ER, endoplasmic reticulum; lb, lipid body; mt, microtubules; r, radicle; v, vacuole; Bars, (**A**) 5 mm, (**B**–**D**) 200 nm.

**Figure 4 cells-09-01778-f004:**
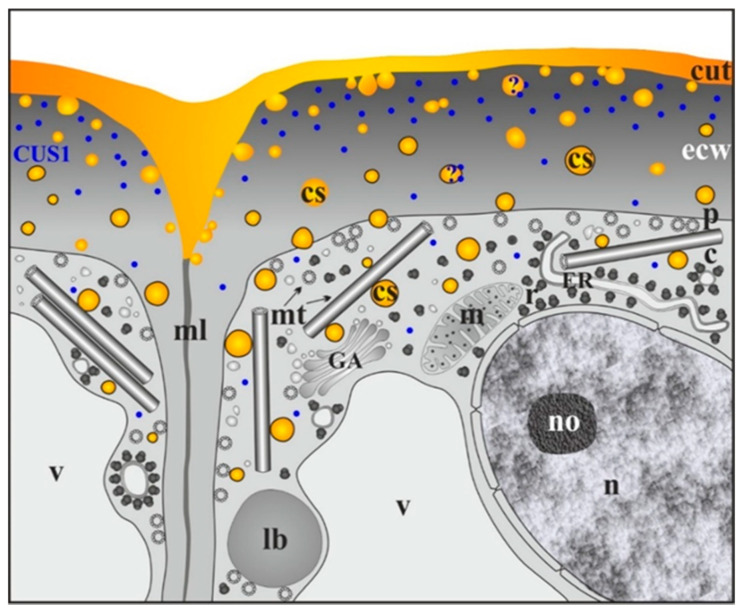
Model of cuticle formation with cutinsome and CUS1 engagement in *S. lycopersicum*. c, cytoplasm; cs, cutinsomes bordered with a shell (black ring) and without it; cut, cuticle; ecw, external cell wall; ER, endoplasmic reticulum; GA, Golgi apparatus; lb, lipid body; m, mitochondrion; ml, middle lamella; mt, microtubule; n, nucleus; no, nucleolus; p, plasmalemma; r, ribosome; v, vacuole; question marks, possible involvement of cutinsomes and CUS1.

**Figure 5 cells-09-01778-f005:**
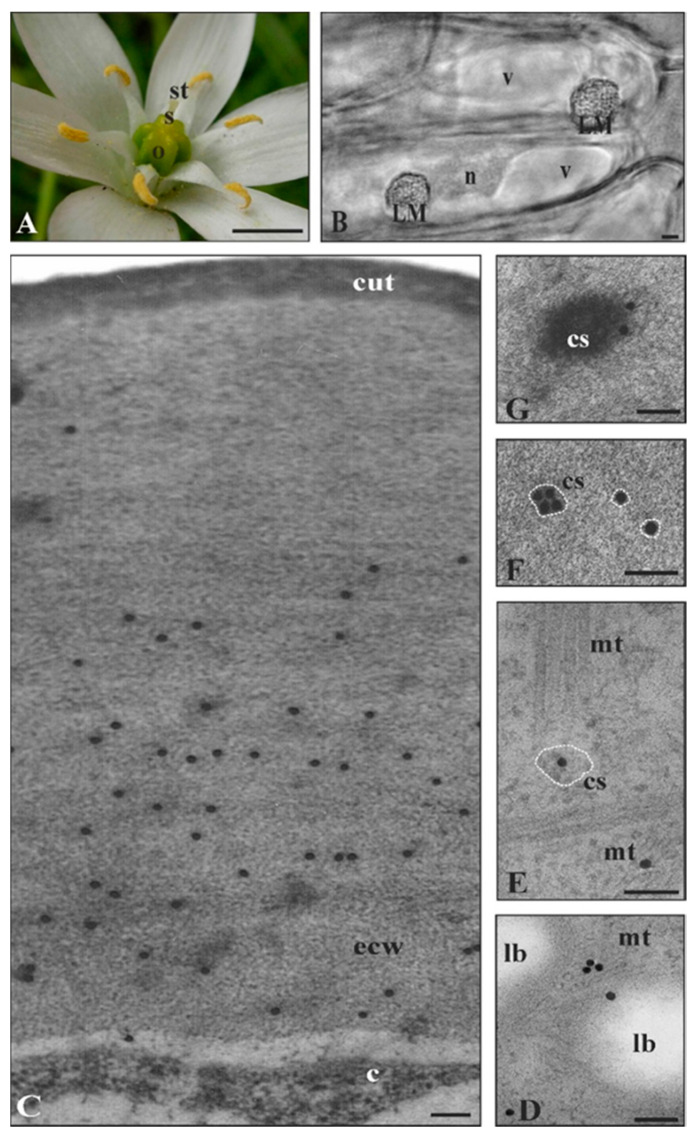
Cutinsome identification and localization in ovary epidermis of *O. umbellatum*. (**A**) Mature flower with clearly visible ovary. (**B**, light microscope image) LMs characteristic of *O. umbellatum* ovary epidermis, in which cutinsomes are formed. Cutinsomes in the (**D**,**E**) cytoplasm and (**C**,**F**,**G**) external cell wall sections (**C**–**G**) fixed in glutaraldehyde and without OsO_4_ after immunogold reaction with anti-cutinsome antibodies. (**G**) Cutinsomes are visible, (rings; **E**,**F**) barely visible, or (**C**,**D**) not visible at all but identified by anti-cutinsome antibodies; c, cytoplasm; cs, cutinsome; cut, cuticle; ecw, external cell wall; lb, lipid body; LM, lipotubuloid metabolon; mt, microtubules; n, nucleus; o, ovary; s, style; st, stigma; v, vacuole; Bars, (**A**) 10 mm, (**B**) 10 µm, (**C**–**G**) 100 nm.

**Figure 6 cells-09-01778-f006:**
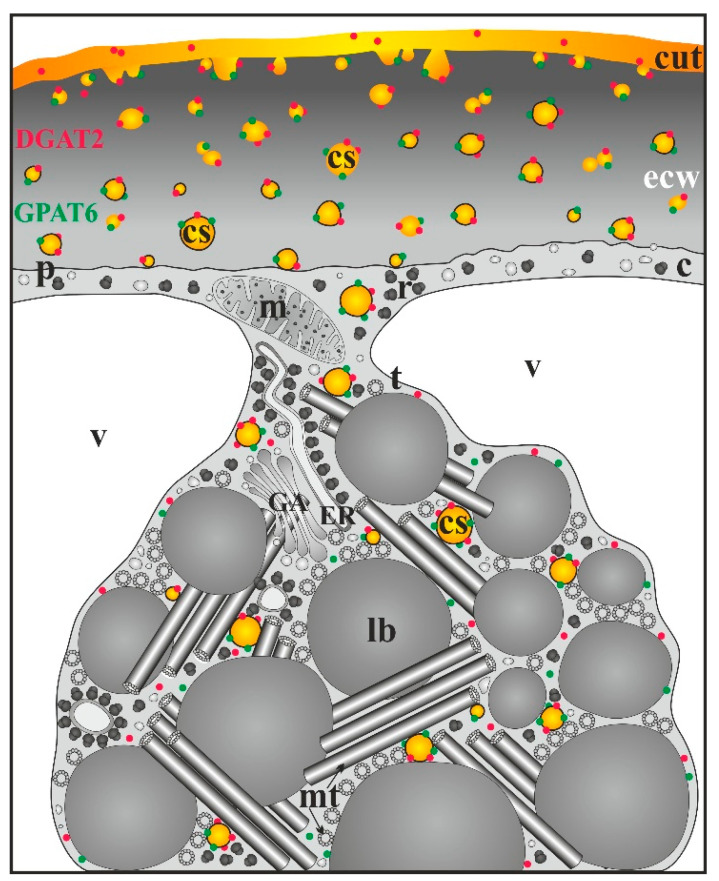
Cuticle formation of *O. umbellatum* ovary with cutinsome engagement and, additionally, with enzyme participation in cutin (GPAT6) and wax (DGAT2) synthesis. These enzymes are transported together with cutinsomes from LM, where their synthesis takes place, to the external cell wall and cuticle, which is their destination. The scheme is on the basis of TEM immunogold studies [30,44]; c, cytoplasm; cs, cutinsome bordered with a shell (black ring) and without it; cut, cuticle; ecw, external cell wall; ER, endoplasmic reticulum; GA, Golgi apparatus; lb, lipid body; m, mitochondrion; mt, microtubules; p, plasmalemma; r, ribosome; t, tonoplast; v, vacuole.

**Figure 7 cells-09-01778-f007:**
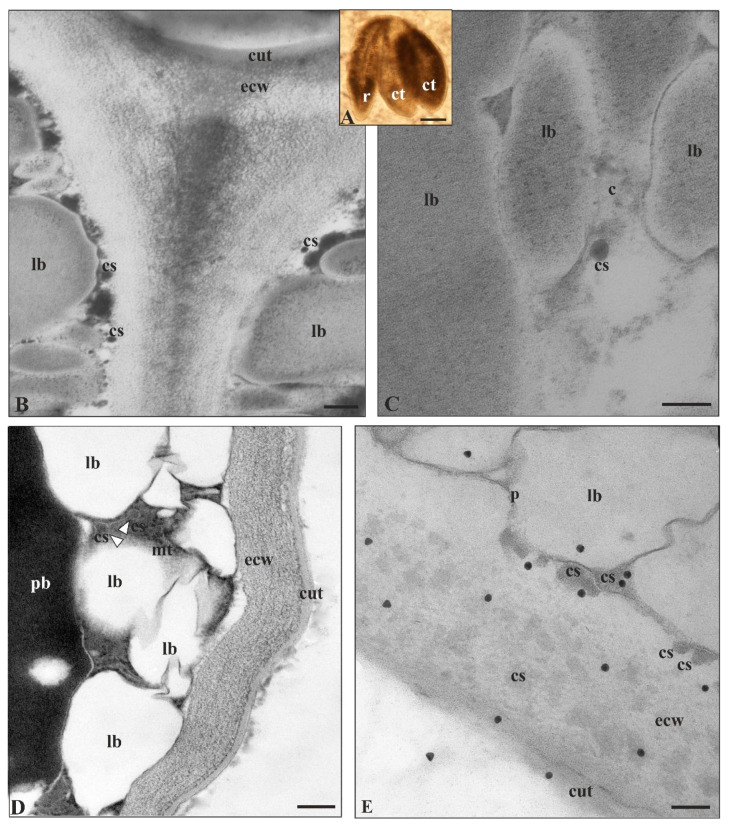
Cutinsomes of *A. thaliana* embryo cotyledon epidermis. (**A**) Isolated mature embryo embedded in resin with well-developed cotyledons (**B**,**D**) whose surface is already covered with clearly visible cuticle. (**B**–**E**) Cutinsomes formed in cytoplasm (**E**) moving to the plasmalemma and external cell wall where they undergo fusion and build cuticle. (**E**) Cutinsomes labeled (20 nm gold grains) after the immunogold reaction using specific antibodies in the external cell wall and cuticle. (**E**) Gold grains are often situated at cutinsomes or at the places where cutinsomes are invisible due to their disintegration, yet their components are still recognized by antibodies [31,66,68]. Moreover, cutinsomes that are visible but not labeled may result from masking their epitopes by other cell wall or cuticle components [84,85,86]. TEM sections fixed (**B**,**C**) with glutaraldehyde and OsO_4_—cutinsomes seen as dark, electron dense structure, and (**D**) without OsO_4_—cutinsomes hardly visible as slightly gray structures (white triangle) against electron-dense cytoplasm; c, cytoplasm; cs, cutinsome; ct, cotyledon; cut, cuticle; ecw, external cell wall; lb, lipid body; mt, microtubule; p, plasmalemma; pb, protein body; r, radicle; Bars, (**A**) 100 µm, (**B**,**D**) 200 nm, (**C**,**E**) 100 nm.

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
