# Peer review of "The Role of Cutinsomes in Plant Cuticle Formation"

_cells, 2020, doi:10.3390/cells9081778_

Round 1
Reviewer 1 Report
Dear authors,
You have modified your first version of the review paper, however, there remain numerous points to be discussed and corrected.
Line 59-73: As you well state in your letter to the reviewers you have performed pioneer studies to lipotuboluid metaboloms and cutinsomes involved in cutin synthesis. However, a reader that is not already knowledgable in cutin biosynthesis cannot recognize this. The paragraph has to be rewritten in a way that information that is based on many different approaches, including mutant analysis and in vivo labeling, and has been already reviewed or is text book knowledge and novel results that are based on a few studies are clearly indicated. As your pioneer research has led to intriguing results, the discussion should highlight which future research should be performed in the future to solidify your results. For example, up to now there is no DGAT gene identified to be involved in cutin synthesis. This writing would better highlight your contribution to the field and not confuse the reader. Recent reviews should be cited for the well-established aspects of cutin biosynthesis, including the review of Fich et al, 2016 that was cited oddly. You cite a paper characterizing oil-rich tubers having no cuticle.
Line 80: BODYGUARD is not a GDSL lipase, but belongs to the a b hydrolase superfamily to which also the GDSL lipases belong.
Line 120 What do you mean with “real” cutinsomes? In planta?
Figure 2
The labeling of the cell wall with the “cutinsome” antibody does not follow any specific pattern thus not well strengthening the hypothesis that it labels a process in cutin formation, i.e. it is not enriched in the cell wall/cuticle interface. It also labels the resin area outside of the plant tissue, thus shows considerable unspecific binding. No information is given that labeling occurred only in the outer epidermal cell walls underneath the cuticle or is specific to cells forming a cuticle, typically the shoot epidermis or is absent/less abundant in mutants with reduced cuticle formation. Such studies would be well feasible.
Figure 3
Although TEM pictures are of high quality a correlation between cutin formation and cutinsomes is not obvious because they are very low in number.
Figure 5
It is understandable that preservation of structure varies with the TEM method used. However, when you can’t see what you label (i.e. “unvisible cutinsomes”) and you have not extensively studied the specificity of the antibody you do not know what you are labeling and cannot distinguish specific labeling from unspecific labeling or of labeling another structure than this which you wish to study. The presence of the label only at the inner part of the cell wall speaks rather for labeling of a structure of the cell wall.
Schematic diagrams:
Cutinsomes are often, but not always displayed a black line around the border. No explanation is given. In TEM no structure is visible that could correlate to a border.
Figure 7:
In contrast to the other TEM pictures, the structures in Figure 7B and 7C are unusually badly preserved for a conventional fixation and can therefore not be interpreted. One cannot distinguish whether one sees plasmolysis or cytoplasm. The label in D is visible at all structures at similar density. As visible from the picture (not well annotated), the embryo was dissected out of the seed. Thus, the label was then also at the resin in D? Specificty of the labeling in D was not good.
Discussion:
In the discussion you list that many researchers found electron-dense structures as cutinsomes but did not identify them correctly. However, the cellular context in which they occur does not always indicate a possible involvement in cutin biosyntehsis. For example, Mc Farlane et al, found many electron-dense vesicles in the outer integument of the seed coat at a time when mycilage was produced and not when the cuticle was layed down. The amount of vesicles is in no relationship to the minor layer of cuticle, thus the observed electron -dense structure and visicles would correlate rather to the massive mycilage production.
Cutinsomes involved in cutin formation should be present only in cells of high cutin biosynthetic acitivity (in case the antibody is specific). Not all electron-dense globular structures are cutinsomes. Please review the literature in respect to the cutin biosynthesis activity of the characterized cells/tissues and remove other citations.
Minor issues:
Glicerol has to be correct to glycerol at many places.
Prospects: Perspectives would be more appropriate
Author Response
RESPONSES TO REVIEWER 1
Authors would like to thank the Reviewer for valuable comments which helped to improve the paper.
The changes in the manuscript were highlighted with the use of "Track Changes" function in Microsoft Word. Number of lines is in accordance with earlier submitted manuscript.
- Line 59-73: As you well state in your letter to the reviewers you have performed pioneer studies to lipotuboluid metaboloms and cutinsomes involved in cutin synthesis. However, a reader that is not already knowledgable in cutin biosynthesis cannot recognize this. The paragraph has to be rewritten in a way that information that is based on many different approaches, including mutant analysis and in vivo labeling, and has been already reviewed or is text book knowledge and novel results that are based on a few studies are clearly indicated.
This paragraph was clarified and simplified in order not to confuse readers. The sentence containing lipotubuloid metabolon (LM) was removed, because this structure belongs to cytoplasm and it is not presented in Fig. 1. However, all knowledge about this cytoplasm structure is included in paragraph 2.2.
As your pioneer research has led to intriguing results, the discussion should highlight which future research should be performed in the future to solidify your results. For example, up to now there is no DGAT gene identified to be involved in cutin synthesis. This writing would better highlight your contribution to the field and not confuse the reader.
The knowledge about cutinsome participation at non-enzymatic pathway of cuticle biosynthesis is still incomplete. The studies have been conducted only on several species so far, so further research is needed to enrich this knowledge. Moreover, it is still unknown how cutinsomes are translocated through a cell wall, that is why we are going to resolve this problem, which was mentioned in Conclusions and Perspectives.
Recent reviews should be cited for the well-established aspects of cutin biosynthesis, including the review of Fich et al, 2016 that was cited oddly. You cite a paper characterizing oil-rich tubers having no cuticle.
The reference Liu et al. 2020 (most probably mistakenly indicated as Fich et al. 2016) was removed.
Information directing a reader to review literature concerning cuticle biosynthesis was added as follows: “Recently, many reviews have appeared that have consolidated the knowledge on the enzymatic pathway of cuticle lipid component biosynthesis, cutin and waxes [3,7,17-19,35-38]. These works present cuticle biosynthesis from different angles and were based on various experimental models, including mutant analysis and in vivo labelling. The current review is devoted to the non-enzymatic pathway of cutin biosynthesis, which has recently been described in literature, and only necessary information on enzymatic cutin biosynthesis is included.”
- Line 80: BODYGUARD is not a GDSL lipase, but belongs to the a b hydrolase superfamily to which also the GDSL lipases belong.
Reviewer 1 is right and we would like to thank for drawing our attention to this issue. This sentence was corrected.
- Line 120 What do you mean with “real” cutinsomes? In planta?
In a phrase ‘real cutinsomes’ a word ‘real’ was indeed unfortunately used so this word was removed.
- Figure 2
The labeling of the cell wall with the “cutinsome” antibody does not follow any specific pattern thus not well strengthening the hypothesis that it labels a process in cutin formation, i.e. it is not enriched in the cell wall/cuticle interface. It also labels the resin area outside of the plant tissue, thus shows considerable unspecific binding. No information is given that labeling occurred only in the outer epidermal cell walls underneath the cuticle or is specific to cells forming a cuticle, typically the shoot epidermis or is absent/less abundant in mutants with reduced cuticle formation. Such studies would be well feasible.
We want to explain that the anti-cutinsome antibodies (used in our immunogold studies), were obtained previously by Dominguez and coworkers and they tested them in terms of specificity; the tests clearly showed that no other wall structures were labelled (Dominguez et al. 2010 [66]). Specificity of these antibodies was also confirmed in other papers (Kwiatkowska et al. 2014, Stępiński et al. 2016, 2017 [70,30,31]). These studies also leave no doubt that no other wall structures are labelled with these antibodies. In the research by Kwiatkowska et al 2014 [70] Fig. 3a. clear specific labelling of external cell wall is also shown.
Moreover, some biological methods (immunofluorescence, autoradiography) leave background - that is why a few traces of labelling can be observed outside of specific labelling (hence the tag in the resin).
The recent manuscript Selgado et al. 2020 [56] shows that a linear relationship between the abundance of cutinsomes and the final amount of cuticle is not necessarily expected. Indeed, a similar situation was observed for BDG mutants (Kurdyukov et al. 2006 [51] and Jacobson et al. 2016 [52]). BDG loss of function mutants showed an increase in the amount of cutin present in fully expanded Arabidopsis leaves. However, when younger stages of development were analyzed, up to 60% reduction in cutin load was observed, leading to the conclusion that in later stages of development other genes compensated the effect of the reduction of cutin due to BDG loss of function.
Cutinsomes were found at the early stages of development and they participated in the synthesis of cutin during the cell division stage (in tomato). Analysis of the CUS1 mutant showed that this gene does not participate in cutin deposition during this early stage of development and as such, the mutant exhibited a similar cutinsome labelling as the control.
- Figure 3
Although TEM pictures are of high quality a correlation between cutin formation and cutinsomes is not obvious because they are very low in number.
The TEM pictures are taken in large magnification, and so they present a small area, whereas in the case of small magnification those small nanostructures can be invisible. That is why it is extremely difficult to find and capture into one image certain structures that actually exist there in greater number. Even if the structures are actually abundant in the given area, still in the case of cross-sections of ultra-thin individual samples you would see just a small number of them or they might not be visible at all. Of course, quite a large number of pictures were observed in TEM where the structures in question occurred, but only the pictures of the highest quality were selected for the final paper.
- Figure 5
It is understandable that preservation of structure varies with the TEM method used. However, when you can’t see what you label (i.e. “unvisible cutinsomes”) and you have not extensively studied the specificity of the antibody you do not know what you are labeling and cannot distinguish specific labeling from unspecific labeling or of labeling another structure than this which you wish to study. The presence of the label only at the inner part of the cell wall speaks rather for labeling of a structure of the cell wall.
In our opinion, the labelling in Fig. 5 is specific as discussed in the paper of Kwiatkowska et al. 2014 [70]. In this work, Table 1 compares labelling densities (number of gold particles per 1 μm2) of cellular compartments of O. umbellatum ovary epidermal with parenchyma cells. Specific labelling in epidermal cells was observed. The labelling density was higher in lipotubuloid metabolon than in the ground cytoplasm and was highest in the external cell wall. In the current review in Fig. 5, labelling is most abundant also in the cell wall in places where the cut structures are visible, which in our opinion are single cutinsomes or fused. These structures getting through the elements of the cell wall further in the direction of the forming cuticle, lose their shell and then may no longer mark. In addition, as demonstrated by Marcus et al. 2008 [86] and Guzman et al. 2014 [87,88], the fact that some elements that are visible in the cell wall but not labelled may result from masking their epitopes by other cell wall or cuticle components [87,88]. Moreover, gold particles connected with the cutinsomes at the border of the cuticular layer and cuticle proper as well as in the cuticle proper were observed by Kwiatkowska et al (2014) [70] and are visible in Fig. 8a, b.
In addition, the explanation of the specificity of the antibodies was given above.
- Schematic diagrams:
Cutinsomes are often, but not always displayed a black line around the border. No explanation is given. In TEM no structure is visible that could correlate to a border.
In the corrected version of the paper, we have improved the descriptions of Figs 1, 4, 6, adding an explanation of the black ring on most cutinsomes, which means shell. It is not always visible on electrograms. Unfortunately, the shell is not visible on the ultrastructural images of the present work, and we did not want to repeat the photos from previous works, however it is presented in the works of Heredia-Guerrero et al. 2008 [63] - Fig. 4A, Heredia-Guerrero et al. 2011[65] - Fig. 2A, Stępiński et al. 2016 [30]- Fig. 1C.
- Figure 7:
In contrast to the other TEM pictures, the structures in Figure 7B and 7C are unusually badly preserved for a conventional fixation and can therefore not be interpreted. One cannot distinguish whether one sees plasmolysis or cytoplasm. The label in D is visible at all structures at similar density. As visible from the picture (not well annotated), the embryo was dissected out of the seed. Thus, the label was then also at the resin in D? Specificty of the labeling in D was not good.
Figure 7 was corrected. Photos 7B, 7C and 7D were exchanged for technically better ones. In Fig 7E, cutinsomes near plasmalemma and in the external cell wall are labelled where they lose their shell and undergo fusion. The description: "Isolated mature embryo embedded in resin with well-developed cotyledons" was added to Fig.7A. In the immunogold technique, labelling is also often observed in resin, but it is scarce. Only one gold grain is visible there.
- Discussion:
In the discussion you list that many researchers found electron-dense structures as cutinsomes but did not identify them correctly. However, the cellular context in which they occur does not always indicate a possible involvement in cutin biosyntehsis. For example, Mc Farlane et al, found many electron-dense vesicles in the outer integument of the seed coat at a time when mycilage was produced and not when the cuticle was layed down. The amount of vesicles is in no relationship to the minor layer of cuticle, thus the observed electron -dense structure and visicles would correlate rather to the massive mycilage production.
Cutinsomes involved in cutin formation should be present only in cells of high cutin biosynthetic acitivity (in case the antibody is specific). Not all electron-dense globular structures are cutinsomes. Please review the literature in respect to the cutin biosynthesis activity of the characterized cells/tissues and remove other citations.
As far as discussion is concerned we want to explain why we cited the work of McFarlane et al. 2008 [88]. In the Figs we saw globular structures, which in our opinion corresponded to the size of cutinsomes, but were actually shown in relation to mucillage seed of A. thaliana. However Reviewer 1 claims that, not all globular structures must be the cutinsomes, answer of course Reviewer 1 is right, but on the other hand, unless you also check them with anti-cutinsome antibodies, you cannot be sure. After all, on the seed integument surface there is initially a cuticle (it is visible in Fig.3B in McFarlane et al. paper but not described). However, since this is not proven, and these are only our presumptions, we removed this fragment and the reference from the corrected manuscript.
Nevertheless, we certainly cannot delete the text about self-assembly of suberin aggregates and creating a suberin layer as well as, similar processes were also implicated in exine development (lines 295-304), that is the key part of this work.
Minor issues:
Glicerol has to be correct to glycerol at many places.
The term glicerol was replaced with glycerol throughout the text.
Prospects: Perspectives would be more appropriate
Conclusions and Prospects was replaced with Conclusions and Perspectives.

Reviewer 2 Report
Good revision and your cartoon diagrams are very helpful. I endorse publication of this.
Author Response
.

Reviewer 3 Report
This manuscript titled “The role of cutinsomes in plant cuticle formation” by Dariusz Stepinski and coauthors is clearly written, well-structured and nicely illustrated. The authors have covered the subject in depth; discussing and summarising our current knowledge of all aspects of cutinsome properties.
Most of the relevant literature from the past two decades is cited judiciously including occasional references to early pioneering work. As a result, this manuscript provides the reader with a very interesting and easy to read overview of this line of research progressing from historical background to latest discoveries.
For these reasons I have no hesitation in recommending this work for publication in Cells.
Comments and editorial corrections:
P2 line 56: Lipid transfer proteins were also reported to be involved in the export of cuticular components (DeBono et al. 2009; Lee et al. 2009b; Kim et al. 2012), but the mechanism underlying their roles in export remains to be determined.
P2 line 62: correct the term glycerol throughout the text.
P10 lines 264: many illustrations in the article are related to the use of anti cutinsome antibodies, it would be useful for the reader if the authors gave more information on these antibodies.
P11 line 313: One aspect that is not really discussed by the authors in the manuscript is the transport of cutinsomes through the cell wall to access the cuticle in formation, is there information in the literature on this process that could be added in this last part of the manuscript ?
Author Response
RESPONSES TO REVIEWER 3
Authors would like to thank the Reviewer for valuable comments which helped to improve the paper.
The changes in the manuscript were highlighted with the use of "Track Changes" function in Microsoft Word. Number of lines is in accordance with earlier submitted manuscript.
P2 line 56: Lipid transfer proteins were also reported to be involved in the export of cuticular components (DeBono et al. 2009; Lee et al. 2009b; Kim et al. 2012), but the mechanism underlying their roles in export remains to be determined.
Sentence “Lipid transfer proteins were also reported to be involved in the export of cuticular components (DeBono et al. 2009; Lee et al. 2009b; Kim et al. 2012), but the mechanism underlying their roles in export remains to be determined” was added (P2 line 58).
P2 line 62: correct the term glycerol throughout the text.
The term glicerol was replaced with glycerol throughout the text.
P10 lines 264: many illustrations in the article are related to the use of anti cutinsome antibodies, it would be useful for the reader if the authors gave more information on these antibodies.
Information on anti-cutinsome antibody was added at the end of Introduction where it was mentioned for the first time (P4 line 116) instead of P10 line 264.
P11 line 313: One aspect that is not really discussed by the authors in the manuscript is the transport of cutinsomes through the cell wall to access the cuticle in formation, is there information in the literature on this process that could be added in this last part of the manuscript ?
As cutinsome participation in cuticle biosynthesis is a relatively new conception, there is no literature on their transport. Transport of cutinsomes via microtubules through cytoplasm has only been observed as was mentioned in the manuscript; thus the mechanism of their translocation through cell wall toward cuticle remains to be determined and this task was also enclosed in perspectives (P11 line 316).

Round 2
Reviewer 1 Report
The authors have replied to all my commments sufficiently well.
This manuscript is a resubmission of an earlier submission. The following is a list of the peer review reports and author responses from that submission.
Round 1
Reviewer 1 Report
The manuscript describes the ways of cuticle formation on the surface of plant organs. It is well written and raises important issues related to the formation of compounds that play an important role in plant organisms. In addition, the work is enriched with figures and diagrams well illustrating the described processes.
Reviewer 2 Report
Dear Authors,
the review is highly speculative and at many places this aspect is not sufficiently highlighted. Other information is incomplete to a degree that it gives an incorrect view (e.g. in the abstract: CUS1 mainly synthetizes linear chains. This is only true in vitro; in contrast analysis of cus1 mutants indicate the contrary, thus the CUS1 enzymatic mechanism has been not yet elucidated). Claims are thus much too broad.
The summary of the cutin biosynthesis pathway is inappropriate (lines 60 -72) as it is based on a few research papers of the corresponding author featuring ultrastructural and histochemical observations in which causality between the observations to cutin formation has been not sufficiently investigated. The evidence that GPAT6 and DGAT2 may also be present in the apoplast be carried by cutinsomes there is only weak (Stepinski, Kwiatkowska et al. 2016). No evidence exists that DGATs plays a role in cutin biosynthesis. The data are too weak to be the basis of a revised view of the cutin biosynthetic pathway.
In contrast, important findings about cutin formation that have been based on many different approaches and methodologies, including analysis of mutants and transgenic plants, in vivo labelling of proteins and structural cutin analysis in situ are not mentioned; as reviewed in (Dominguez, Heredia-Guerrero et al. 2015, Fich, Segerson et al. 2016) and listed below.
In the following the attractive diagrams are more based on ideas than on data, therefore they are dangerously misleading to the reader who will be attracted to used them in the future disconnected from the context.
A few examples of research data that are not considered:
GPATs use acyl-CoA bound fatty acids as substrate that is not present in the large amounts-if at all- in the apoplast (Yang, Simpson et al. 2012). GPATs are ER-bound (Gidda, Shockey et al. 2009).
The authors give only an incomplete view about CUS1 function. Labeling of cutin in cus1 mutants in situ showed that CUS1-knockout affects cutin branching are not mentioned (Philippe, Gaillard et al. 2016). CUS1 is the only cutin-forming enzyme mentioned even when CUS1 belongs to a large gene family of which many members are expressed in the epidermis during cutin formation and are likely also involved in cutin formation (Yeats, Huang et al. 2014, Fernandez-Pozo, Zheng et al. 2017).
Other enzymes that are involved in cutin formation are not mentioned at all, such as BDG and DCR (Kurdyukov, Faust et al. 2006, Panikashvili, Shi et al. 2009, Jakobson, Lindgren et al. 2016); even when DCR has been shown to form trihydroxyacylglycerols in vitro (Rani, Krishna et al. 2010), which might support the existence of cutinsomes; but not the localization of GPAT6 and DGAT in the apoplast since DCR acting downstream is localized in the cytoplasm (Panikashvili, Shi et al. 2009).
Dominguez, E., et al. (2015). "Plant cutin genesis: unanswered questions." Trends in Plant Science 20(9): 551-558.
Fernandez-Pozo, N., et al. (2017). "The Tomato Expression Atlas." Bioinformatics 33(15): 2397-2398.
Fich, E. A., et al. (2016). The Plant Polyester Cutin: Biosynthesis, Structure, and Biological Roles. Annual Review of Plant Biology, Vol 67. S. S. Merchant. 67: 207-233.
Gidda, S. K., et al. (2009). "Arabidopsis thaliana GPAT8 and GPAT9 are localized to the ER and possess distinct ER retrieval signals: Functional divergence of the dilysine ER retrieval motif in plant cells." Plant Physiology and Biochemistry 47(10): 867-879.
Jakobson, L., et al. (2016). "BODYGUARD is required for the biosynthesis of cutin in Arabidopsis." New Phytologist 211(2): 614-626.
Kurdyukov, S., et al. (2006). "The epidermis-specific extracellular BODYGUARD controls cuticle development and morphogenesis in Arabidopsis." Plant Cell 18(2): 321-339.
Panikashvili, D., et al. (2009). "The Arabidopsis DCR encoding a soluble BAHD acyltransferase is required for cutin polyester formation and seed hydration properties." Plant Physiology 151(4): 1773-1789.
Philippe, G., et al. (2016). "Ester Cross-Link Profiling of the Cutin Polymer of Wild-Type and Cutin Synthase Tomato Mutants Highlights Different Mechanisms of Polymerization." Plant Physiology 170(2): 807-820.
Rani, S. H., et al. (2010). "DEFECTIVE IN CUTICULAR RIDGES (DCR) of Arabidopsis thaliana, a gene associated with surface cutin formation, encodes a soluble diacylglycerol acyltransferase." Journal of Biological Chemistry 285(49): 38337-38347.
Stepinski, D., et al. (2016). "Cutinsomes and cuticle enzymes GPAT6 and DGAT2 seem to,travel together from a lipotubuloid metabolon (LM) to extracellular matrix of O. umbellatum ovary epidermis." Micron 85: 51-57.
Yang, W. L., et al. (2012). "A Land-Plant-Specific Glycerol-3-Phosphate Acyltransferase Family in Arabidopsis: Substrate Specificity, sn-2 Preference, and Evolution." Plant Physiology 160(2): 638-652.
Yeats, T. H., et al. (2014). "Tomato Cutin Deficient 1 (CD1) and putative orthologs comprise an ancient family of cutin synthase-like (CUS) proteins that are conserved among land plants." Plant Journal 77(5): 667-675.

Reviewer 3 Report
This is a unique and very interesting manuscript. The authors synthesize the limited data from the literature concerning the deposition mechanism of lipids into the cuticle of several plants. This is not an easy task as structures like cutisomes has been difficult to characterize in primary research. The manuscript provided an up to date summary of what is known and supplemented it with fine diagrams and micrographs.
There are few minor areas of concern: Fig. 3b does not appear to be in focus; lines 304-309 should be re-written as the evolutionary "story" requires clarification- what data is there concerning cutie in charophytes.
Overall this is a valuable manuscript.